# ViFu: Visible Part Fusion for Multiple Scene Radiance Fields

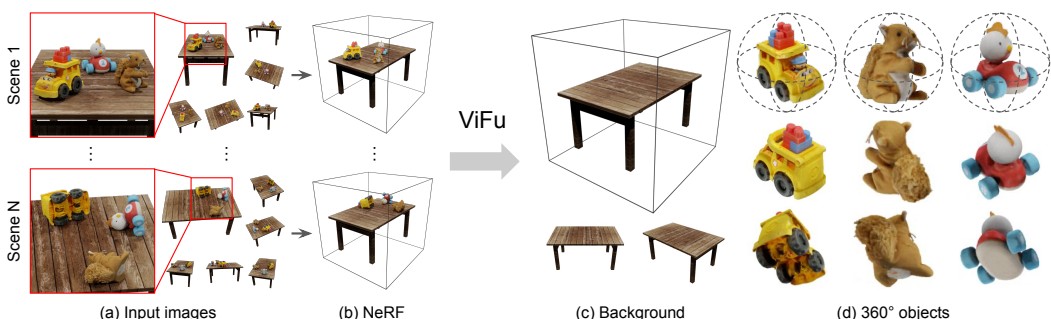

Figure 1: **An overview of our approach**. From (a) multi-view images of multiple scenes with different object placements, ViFu recovers the appearance and 3D geometry of (c) clean static backgrounds and (d) 360° foreground objects. Radiance fields representation supports free-view rendering of the recovered background scene and foreground objects.

## Abstract

In this paper, we propose a method to segment and recover a static, clean background and 360° objects from multiple scene observations. Recent works have used neural radiance fields to model 3D scenes and improved the quality of novel view synthesis, while few studies have focused on modeling the invisible or occluded parts of the training images. These under-modeled parts constrain both scene editing and rendering view selection. Our basic idea is that, by observing the same set of objects in various arrangement, so that parts that are invisible in one scene may become visible in others. By fusing the visible parts from each scene, occlusion-free rendering of both background scene and foreground objects can be achieved.

We decompose the multi-scene fusion task into two main components: (1) objects/background segmentation and alignment, where we leverage point cloud-based methods tailored to our novel problem formulation; (2) radiance fields fusion, where we introduce *visibility field* to quantify the visible information of radiance fields, and propose *visibility-aware rendering* for multiple scene fusion, ultimately obtaining clean background and 360° object rendering. Comprehensive experiments were conducted on synthetic and real datasets, and the results demonstrate the effectiveness of our method.

The code will be release for research purposes upon paper acceptance.

## 1 Introduction

Recently, the advance of neural rendering with implicit representation has received attention for its numerous real-world applications, including virtual reality, games, movies, and more. One of the pioneering works is neural radiance field (NeRF) Mildenhall et al. (2020), which uses a neural network to model the 3D space as a continuous radiance field, enabling the reconstruction of the detailed geometry and appearance of a scene from multi-view images. There has been significant follow-up work to explore the extension of NeRF in the direction of fast optimization Yu et al. (2021); Müller et al. (2022); Chen et al. (2022); Sun et al. (2021), generalization Yu et al. (2020);

Wang et al. (2021b), dynamic scenes Park et al. (2020); Tretschk et al. (2020); Pumarola et al. (2020), human body Xu et al. (2022); Peng et al. (2021); Noguchi et al. (2022b) or articulated objects Noguchi et al. (2021; 2022a) modeling, appearance editing Liu et al. (2021); Kobayashi et al. (2022), shape editing Xu & Harada (2022); Yuan et al. (2022), etc.

In particular, compositional scene modeling is one of the popular directions in which individual parts of a scene, such as a background scene or foreground objects, are independently modeled rather than treating the entire scene as a whole. It represents the whole scene as a composition of background scene and foreground objects, enabling applications such as scene segmentation Zhi et al. (2021), object movement or removal Yang et al. (2021); Wu et al. (2022), independent object rendering Jang & de Agapito (2021), etc.

While an increasing number of works have attempted to use NeRF for compositional scene modeling, an obvious but challenging issue has been left unaddressed: background or objects occluding each other can result in parts of the scene that cannot be observed from the training images, thereby causing under-modeled parts in the scene. As a result, the movement/removal of objects, or rendering from certain viewpoints can expose these under-modeled parts, leading to poor rendering results with artifacts (*e.g.*, Fig. 2 (c)). Especially in tasks requiring clean backgrounds or manipulation of object placement, such as indoor scene reconstruction or robotics applications, this issue becomes particularly pronounced. Specifically, we consider two cases of under-modeling: (1) under-modeled background scene, such as the desktop, where the contact surface with the foreground objects is invisible during training, leading to artifacts when removing or moving foreground objects; (2) under-modeled foreground objects, where the invisible surface is exposed when rendering with changing the object's pose (*e.g.*, laying it down), causing artifacts. To the best of our knowledge, no previous studies have attempted to address these issues.

In this work, we explore compositional scene modeling from the perspective of recovering clean backgrounds and 360° objects. Recovering the above unseen parts from a single scene is challenging and laborious, as it usually requires a hand-designed or learned scene prior, as in image completion tasks. Instead of a single scene, we consider a set of scenes where the background remains static while objects are placed in different positions and poses. Here, the object placement satisfies two conditions: (1) there is no part of the *background* that is always occluded by the object in all scenes, and similarly (2) there is no part of the *objects* surface that is invisible in all scenes (*i.e.*, every part of the background/objects is visible in at least one scene). These two conditions correspond to the two under-modeled cases above, and this multi-scene setup ensures that we have enough information to recover the geometry and appearance of clean background and 360° objects.

Recall that the above key issues come from the invisible part caused by occlusion. To address this issue, given the volumetric nature of the radiance field, we propose *visibility field*, a volumetric representation for quantifying the visibility in scenes. With the proposed visibility field, we compare the visibility of the corresponding part across multiple scenes and fuse the parts with higher visibility to achieve clean background and 360° objects rendering. We dub our proposed idea of visible part fusion as *ViFu*. The basic idea of ViFu is shown in Fig. 2. Furthermore, we leverage the multi-scene setting and propose a method for segmenting objects and backgrounds by exploiting the differences in object placement across each scene. Our segmentation approach is based on the geometric differences w.r.t. clean backgrounds obtained via fusion, which is computationally efficient and simple, and does not require any pre-trained 3D segmentation model.

To verify the effectiveness of ViFu, we created several sets of synthetic scenes containing various objects. We observe that ViFu automatically and accurately segments the background and each object, and achieves pleasing recovery of clean backgrounds and free-view rendering of 360° foreground objects. We also captured videos to create a set of real-world datasets, and the experimental results show that the proposed method also gives promising results for real-world scenes.

In summary, our main contributions are listed as follows:

- We studied the under-modeled invisible parts of NeRF and introduced the setting of complementing the invisible parts by fusing multiple scene information.

- We introduce *visibility field*, a volumetric representation to quantify the visibility of scenes, and propose novel *visibility-aware rendering*, which leverages the visibility field to achieve the fusion of visible parts of multiple scenes.

- We created synthetic and real datasets to validate our idea, and the experimental results show the effectiveness of the proposed method.

## 2 RELATED WORK

**Neural radiance field revisited.** Recently, neural rendering with implicit representations has received significant attention due to its detailed representation of the geometry and appearance of the scene Sitzmann et al. (2019); Yariv et al. (2020); Mildenhall et al. (2020). The most representative work is neural radiance field (NeRF) Mildenhall et al. (2020), which uses neural networks to model the scene as a continuous mapping from position and view direction to radiance color and volume density, enabling geometric and appearance reconstruction and photorealistic novel view rendering. Several follow-up works have been proposed to improve the foundation of NeRF, enabling fast optimization Yu et al. (2021); Müller et al. (2022); Chen et al. (2022); Sun et al. (2021), appearance decoupling Verbin et al. (2021), dynamic scene modeling Pumarola et al. (2020); Park et al. (2020); Tretschk et al. (2020), and more. Nevertheless, these methods have limitations as they model the scene as a whole and do not allow for segmentation or editing of specific parts of the scene.

**Object-centric scene representation.** A new category of object-centric modeling methods has been proposed to enhance the reasoning and editing capabilities of scenes. Specifically, compositional scene modeling methods Zhang et al. (2020); Guo et al. (2020); Niemeyer & Geiger (2021); Wang et al. (2021c); Zhang et al. (2021); Wu et al. (2022) regard the entire scene as a mixture of background and foreground objects, facilitating object-level scene understanding; some methods encode semantic information into scenes, enabling feature-based object query or segmentation Zhi et al. (2021); Wang et al. (2022; 2021a). Another direction explores object-level manipulations on scene content, enabling editing to object appearance Liu et al. (2021); Bao et al. (2023) or geometry Xu & Harada (2022); Yuan et al. (2022). These advancements have made notable progress in manipulating NeRF-based representations, however, our primary concern is that manipulating the original scenes (*i.e.*, object movement or deformation) can inadvertently expose unseen parts and thus lead to artifacts.

**Scene completion for radiance fields.** To address the issue of under-modeled parts being exposed, recent studies have approached it as a 3D inpainting problem and proposed solutions for radiance field representations. NeRF-In Liu et al. (2022) uses masks to segment the foreground objects and performs inpainting to obtain an unoccluded background, while SPin-NeRF Mirzaei et al. (2022) improves on this by introducing the concept of perceptual inpainting to enhance the rendering results. However, these methods only consider the completion of the background part and do not address the invisible parts of the objects. Furthermore, the shadows cast by the original objects still appear as noticeable artifacts in the resulting inpainted regions Liu et al. (2022); Mirzaei et al. (2022).

**Scene fusion for radiance fields.** Some recent works also attempt to fuse NeRF, such as NeRFusion Zhang et al. (2022) or NeRFuser Fang et al. (2023). The objective of these methods is to integrate the individual 3D representations of various local components within a vast scene, thereby obtaining a comprehensive scene rendering. Hence, the main focus of these methods lies in modeling large-scale scenes effectively. Conversely, our approach is centered around addressing occlusions caused by objects within the scene, aiming to reconstruct an occlusion-free background scene and 360° foreground objects by leveraging the visible parts across different scenes.

## 3 METHOD

Consider a static background scene and $M \geq 1$ foreground objects that are placed in different positions and poses resulting in $N \geq 2$ different scenes (*e.g.*, Fig. 1 (a)). For each scene, we capture $L_i$ multi-view images $\{\mathcal{I}_l\}$ and run the structure-from-motion method independently for each scene to obtain the camera parameters (intrinsics and extrinsics) $\{\mathcal{C}_l\}$, where $i \in \{1, ..., N\}$ denotes scene index and $l \in \{1, ..., L_i\}$ denotes camera index of scene $i$. From the calibrated multi-view images, we optimize neural radiance fields (NeRF) $\{\mathcal{S}_i\}$ for each scene. Radiance field is an implicit scene representation that maps spatial position $\mathbf{x} \in \mathbb{R}^3$ and view direction $\mathbf{d} \in \mathbb{S}^2$ to radiance color $\mathbf{c} = (r, g, b)$ and volume density $\sigma$ as $\mathcal{S} : (\mathbf{x}, \mathbf{d}) \mapsto (\mathbf{c}, \sigma)$.

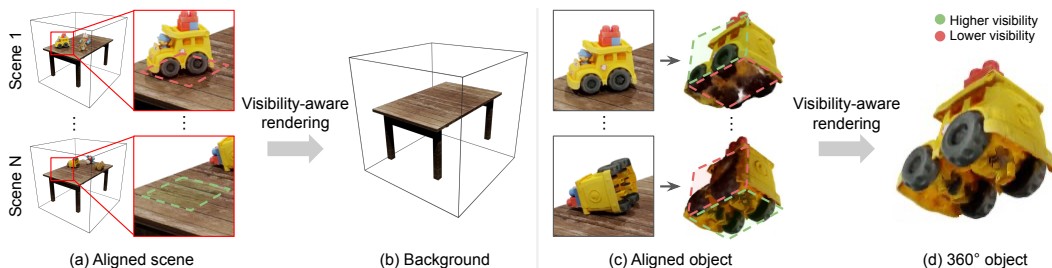

Figure 2: **The basic idea of ViFu**. With pre-computed scene/objects alignment, we compare the visibility of the corresponding parts using the proposed visibility field, and fuse the higher visibility parts of each scene to form the clean background and 360° objects. The details of visibility-aware rendering are shown in Fig. 3.

Our method takes $N$ optimized radiance fields $\{\mathcal{S}_i\}$ as input, automatically splits the scenes into a static background and $M$ foreground objects, and recovers a non-occluded background scene and 360° objects that can be seen from arbitrary view point.

**Assumption.** For our problem formulation, we make the following two ideal assumptions: (1) diverse object *positions*: this ensures visibility of the background scene, implying that every part of the background scene is observable in at least one scene (*i.e.*, no permanently occluded regions); this also facilitates the segmentation of foreground objects, as will be introduced in Sec. 3.2. (2) diverse object *poses*: this guarantees that every part of the object's surface is observable in at least one scene (*e.g.*, no permanently facing-down surfaces).

The assumptions are natural for household objects in everyday scenes: static objects that remain unchanged, such as refrigerators or tables, are considered part of the background; while objects that are frequently moved, such as the toys in Fig. 1, are treated as foreground objects.

### 3.1 METHOD OVERVIEW

Our objective is to perform background/foreground segmentation from multiple scenes and obtain a clean background and 360° objects via fusion. In the general context of 3D modeling, this process can be divided into two main steps: the first involves *internal scene* reasoning, specifically the segmentation of background/foreground within each scene; the second entails *inter-scene* reasoning, which involves matching the segmented background and individual objects among different scenes (*i.e.*, pose alignment for background scene and foreground objects), and subsequently accomplishing the final fusion.

In the following section, we introduce our solutions, specifically tailored for the recent 3D representation of the radiance field. To be more precise, we leverage a point cloud-based approach to perform scene segmentation and alignment (Sec. 3.2), and introduce a novel measure for quantifying the visibility of the radiance fields (Sec. 3.3), which is used in the proposed scene fusion method (Sec. 3.4).

### 3.2 OBJECT SEGMENTATION AND ALIGNMENT

The first step involves background/foreground segmentation and obtaining the relative poses of foreground objects and background scene within each scene. This allows us to align them to their respective common coordinate systems, which are utilized for subsequent fusion purposes (See Fig. 2). For the segmentation and alignment of the radiance fields, we found that existing point cloud-based methods already yield satisfactory results. For simplicity, we introduce here the minimal segmentation and alignment techniques below; however, other more advanced alternatives can also be employed.

We provide a high-level overview of the entire process here, with specific calculations detailed in the supplementary materials. First, we employ Marching Cubes Lorensen & Cline (1987) to convert the radiance field of each scene into a mesh, from which we extract point clouds by surface

sampling. While the placement of individual foreground objects may vary, a substantial overlap of point clouds belonging to the static background scene is sufficient for achieving inter-scene pose alignment through point cloud registration algorithms. Based on the derived relative poses, we utilize the method outlined in Sec. 3.4 to obtain the fused clean background scene, and from which we similarly extract the point cloud corresponding to the background scene. By comparing the differences between the point clouds of each scene and the clean background scene, we can obtain all the point clouds that belong to the foreground objects. Subsequently, a point cloud clustering algorithm allows us to obtain point clouds that belong to each individual foreground object separately. Finally, for each foreground object across scenes, the Hungarian matching algorithm and point cloud registration techniques are used to determine their correspondences and relative poses $\{T_{i,j}\}$. Here $j \in \{1, ..., M\}$ denotes the object index.

### 3.3 VISIBILITY FIELD: QUANTIFYING VISIBILITY IN RADIANCE FIELD

Visibility is an important measure to utilize the *visible part* information across scenes. To quantify the visibility information in the radiance field, we propose *visibility field*, a volumetric representation that maps a 3D position to a scalar-valued visibility:

$$v = v(\mathbf{x}) : \mathbb{R}^3 \to [0, 1]. \tag{1}$$

The proposed visibility $v(\mathbf{x}) \in [0, 1]$ is defined as the proportion of cameras that can observe point $\mathbf{x}$ among all training cameras. Formally, we say that $\mathbf{x}$ can be observed by the camera $\mathcal{C}_l$ means that (1) the projection of $\mathbf{x}$ falls within the interior of the image plane and (2) there is no occlusion between $\mathbf{x}$ and the camera position $\mathbf{o}_l \in \mathbb{R}^3$. For the condition (2), we use the pseudo-depth of the radiance field to determine whether there is occlusion. Specifically, we cast a ray from the camera position $\mathbf{o}_l$ to $\mathbf{x}$ and compute the pseudo-depth $\hat{d}_l$ by volume rendering, and then compare it with the distance from the camera position to the point $d_l = \|\mathbf{x} - \mathbf{o}_l\|$. For camera $\mathcal{C}_l$, we use a binary-valued function $V_l(\mathbf{x}) \in \{0, 1\}$ to denote whether $\mathbf{x}$ can be observed by that camera. If $d_l < \hat{d}_l$, this means that $\mathbf{x}$ is between the object surface and the camera position, thus there is no occlusion, *i.e.*, $V_l(\mathbf{x}) = 1$, otherwise $V_l(\mathbf{x}) = 0$. Considering all training cameras, the visibility of position $\mathbf{x}$ can be computed as:

$$v(\mathbf{x}) = \frac{1}{L} \sum_{l=1}^{L} V_l(\mathbf{x}). \tag{2}$$

Note that the visibility field is independent for each scene and we compute it for all scenes.

### 3.4 VISIBILITY-AWARE RENDERING

We propose *visibility-aware rendering*, a method that obtains occlusion-free rendering by comparing the visibility across multiple scenes. We take the rendering of the clean background to explain its basic idea (Fig. 3 (Left)).

**Background scene.** The first step in comparing scenes is to set them under the same coordinates. Recall that we have obtained the relative pose between the background scenes through point cloud registration in Sec. 3.2. Without loss of generality, we take the first scene ($i = 1$) as a reference and align the scenes $i = 2, ..., N$ to the coordinate system of the first scene. Given that all scenes are aligned to the reference scene, we introduce visibility-aware rendering for the background scene. An illustration is shown in Fig. 3 (Left). For the sample point $\mathbf{x}$ in volume rendering, the proposed visibility-aware rendering, in addition to color and density, also computes the visibility of the sample point in each scene, *i.e.*, $\{\mathbf{c}_i(\mathbf{x})\}$, $\{\sigma_i(\mathbf{x})\}$ and $\{v_i(\mathbf{x})\}$. The idea of visibility-aware rendering is simple: blend the color and density

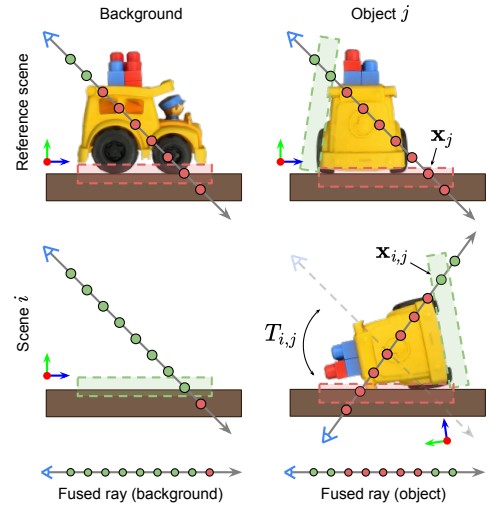

Figure 3: Illustration of visibility-aware rendering in 2D. The colors correspond to higher/lower visibility as shown in Fig. 2.

in each scene according to visibility. Using $w_i$ which satisfy $\sum_i w_i = 1$ to denote the weight of each scene, blended radiance color and volume density can be written as:

$$\hat{\mathbf{c}}(\mathbf{x}) = \sum_{i=1}^{N} w_i(\mathbf{x})\mathbf{c}_i(\mathbf{x}), \; \hat{\sigma}(\mathbf{x}) = \sum_{i=1}^{N} w_i(\mathbf{x})\sigma_i(\mathbf{x}), \tag{3}$$

where $w_i$ is a weight function calculated from visibility that satisfies $\sum_i w_i = 1$:

$$w_i(\mathbf{x}) = \frac{v_i^p(\mathbf{x})}{\sum_{i=1}^{N} v_i^p(\mathbf{x})}. \tag{4}$$

Here $p$ is a hyper-parameter that controls the weights, the larger $p$ is, the greater the contribution of the scene with the highest visibility; and when $p \to \infty$, the above is equivalent to the max-selection function. For simplicity, Fig. 3 shows the case based on max-selection.

The motivation behind the above calculation is to select the parts with less occlusion (*i.e.*, higher visibility) in each scene, and fuse them into the final scene. As a result, volume rendering of the blended radiance color and volume density obtained from Eq. 3 yields a clean background scene, as shown in Fig. 2 (b).

**Foreground objects.** The core idea of visibility-aware rendering for $360°$ objects is basically the same as that for background scene. Similarly, we take the coordinate systems of the foreground objects in scene $i = 1$ as a reference. For foreground object $j$, we denote the position and view direction of the sampled point under the reference coordinate system as $\mathbf{x}_j$, $\mathbf{d}_j$, respectively. For scenes of $i \geq 2$, we use the computed object poses to calculate the corresponding positions $\mathbf{x}_{i,j}$ and view directions $\mathbf{d}_{i,j}$ in each scene as:

$$\mathbf{x}_{i,j} = R_{i,j}\mathbf{x}_j + t_{i,j}, \; \mathbf{d}_{i,j} = R_{i,j}\mathbf{d}_j, \tag{5}$$

where $R_{i,j}$ and $t_{i,j}$ are rotation and translation terms of object poses $T_{i,j} \in \mathrm{SE}(3)$ obtained from Sec. 3.2. Here, $\mathbf{x}_{i,j}$ in fact represents the corresponding point of $\mathbf{x}_j$ in the coordinate system of scene $i$, as shown in Fig. 3 (Left). Then, the blended radiance color and volume density of Eq. 3 for foreground object rendering can be rewritten as:

$$\tilde{\mathbf{c}}(\mathbf{x}_j) = \sum_{i=1}^{N} w_i(\mathbf{x}_{i,j})\mathbf{c}_i(\mathbf{x}_{i,j}), \; \tilde{\sigma}(\mathbf{x}_j) = \sum_{i=1}^{N} w_i(\mathbf{x}_{i,j})\sigma_i(\mathbf{x}_{i,j}). \tag{6}$$

Volume rendering the fusion results obtained from Eq. 6 yields occlusion-free $360°$ foreground objects, as shown in Fig. 2 (d).

Our proposed visibility-aware rendering, despite its simplicity, reasonably achieves the visible part fusion of radiance fields. It's noteworthy that our method share the same paradigm for both background/foreground parts, accomplishing the reconstruction of a clean background scene and $360°$ foreground objects.

## 4 EXPERIMENTS

### 4.1 DATASETS

**Blender synthetic datasets.** We created synthetic datasets using Blender Community (2018). The tables used as background are taken from free 3D models available online. For the foreground objects, we use 3D models from Google Scanned Objects dataset Downs et al. (2022), which contains $360°$ scans of common household objects. We created $N = 3$ sets of scenes, in which foreground objects are under difference placement to ensure that every part of the table and object surfaces is visible in at least one scene. We applied different lighting conditions (uniform light, spotlight, etc.) to test the effectiveness of our method in different environments. We randomly sample camera positions on the hemisphere and render $L = 100$ images for the radiance field optimization. Examples of the synthetic scenes are shown in Fig. 4 (a).

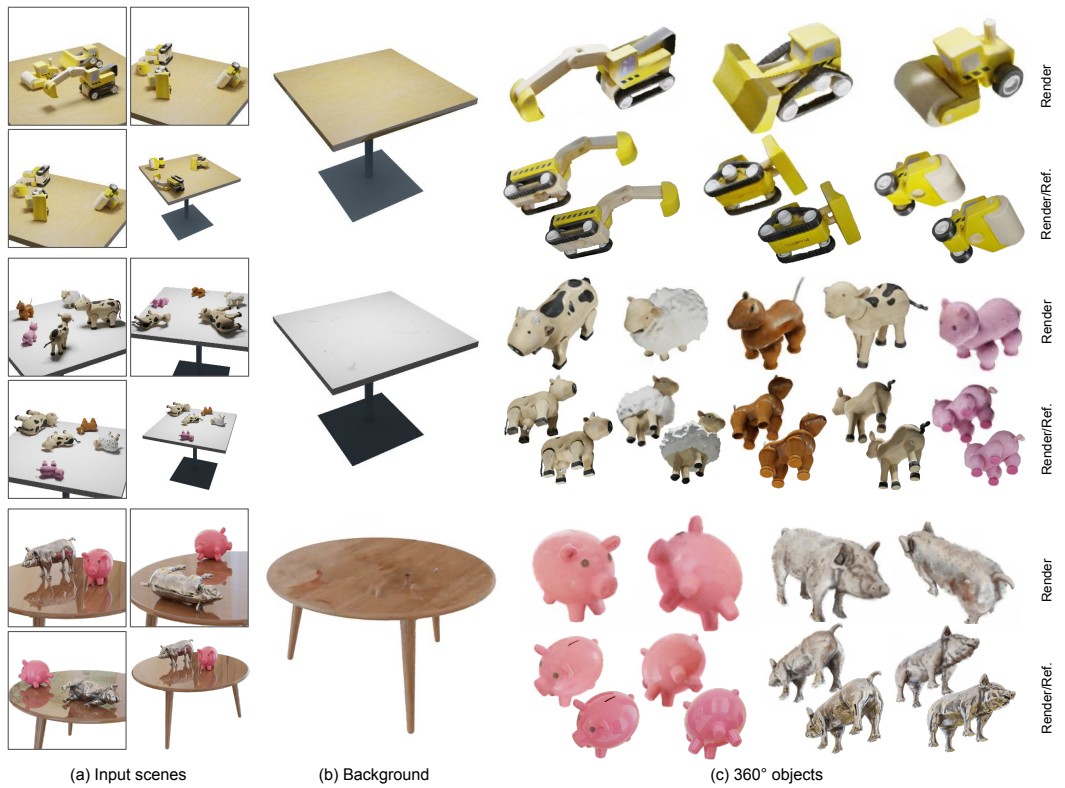

(a) Input scenes      (b) Background      (c) 360° objects

Figure 4: **Results on Blender synthetic datasets**. For pairwise comparisons of foreground objects, the top-left image shows the rendering result of the proposed method, while the bottom-right image shows the reference image (ground truth).

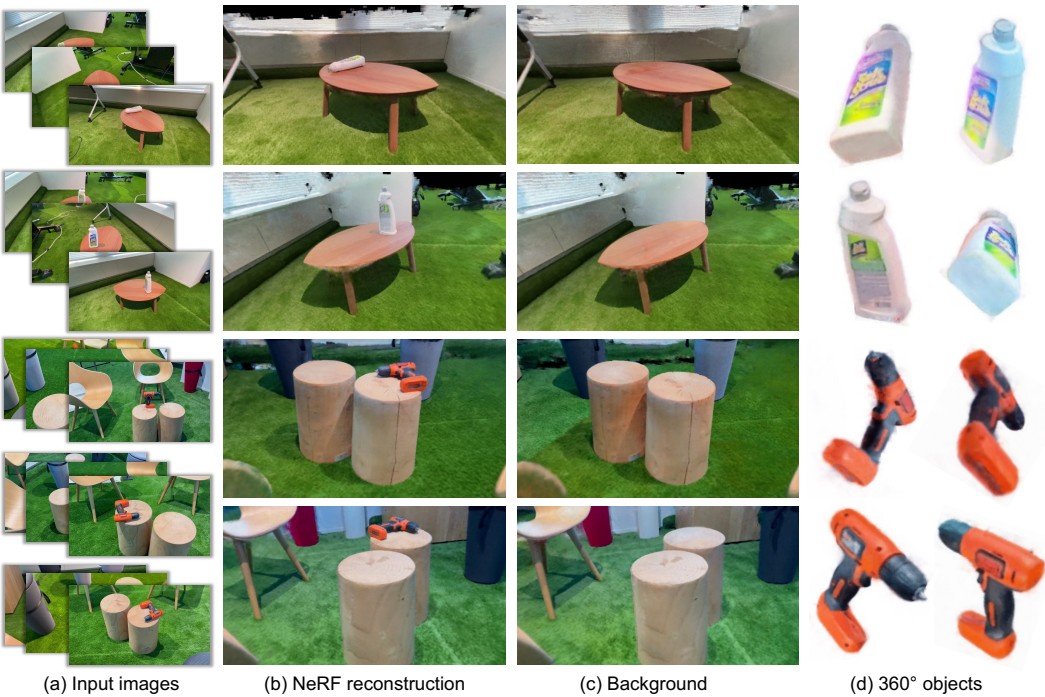

(a) Input images     (b) NeRF reconstruction     (c) Background     (d) 360° objects

Figure 5: **Results on real capture datasets**. (c) and (d) are obtained using the proposed method.

**Real capture datasets.** We created real-world capture datasets to demonstrate the effectiveness of our approach on real datasets. We utilized YCB objects Calli et al. (2015) and created $N = 2$ (for bleach cleanser) or $N = 3$ (for power drill) scenes by placing objects in different configurations. For each scene, we captured a video around it and extracted 60-80 frames, then applied COLMAP Schönberger & Frahm (2016); Schönberger et al. (2016) to obtain the corresponding camera parameters registration. Examples of the real capture scenes are shown in Fig. 5 (a).

## 4.2 RESULTS

We show the qualitative results of Blender synthetic datasets and real capture datasets on Fig. 4 and Fig. 5, respectively. With multiple input scenes, our method can automatically recover a clean background scene and 360° foreground objects.

## 4.3 ABLATION STUDIES

**Impact of light conditions.** We created scenes under three distinct lighting conditions: outdoor environment mapping, indoor environment mapping, and a single point light source. For the background, despite obtaining an acceptable clean background, there exists a certain degree of artifacts due to the presence of shadows or reflections. For objects, certain discontinuities arise due to abrupt changes in lighting conditions or the inherent glossiness of the objects (*e.g.*, pink pig). Additionally, the fused results show a lack of glossiness, suggesting that even for significantly different lighting conditions or glossy objects, our fusion method can neutralize the view-dependent term, yielding appearances close to diffuse colors, which is typically desirable in the context of 360° object reconstruction.

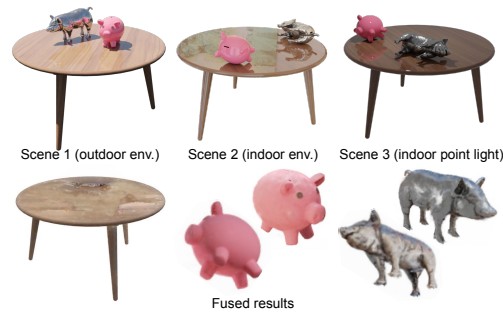

Scene 1 (outdoor env.)    Scene 2 (indoor env.)    Scene 3 (indoor point light)

Fused results

**Impact of weight function.** We study the impact of the hyper-parameter $p$ (exponent of visibility in weight function Eq. 4). We observe that when $p$ is relatively small (*i.e.*, $p = 4$), the results tend to blend color and density more smoothly for each scene. The appearance changes smoothly for foreground objects, however, it also blends the background and non-background (*i.e.*, empty space) parts around them, resulting in a cloud-like artifact. When $p \to \infty$, visibility aware rendering selects the color and density of the scene with the highest visibility as the result of the fusion, and such a max-selection brings discontinuous changes, resulting in sharp changes in the appearance. We observe that $p = 8 \sim 32$ is the appropriate value to obtain continuous appearance interpolation without cloud-like artifacts.

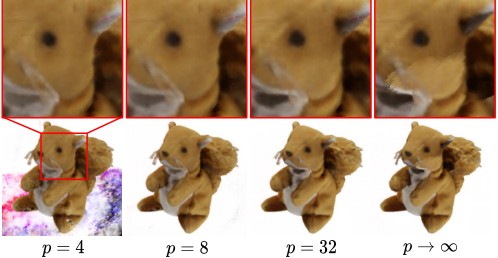

$p = 4$     $p = 8$     $p = 32$     $p \to \infty$

**Impact of the number of scenes.** Right image shows the impact on the results for different numbers of scenes $N$. For $N = 1$, we manually compute the bounding boxes for the background and foreground objects from the point cloud and rendered only the original scene within them. In this case, the invisible parts are not optimized, leading to artifacts in the rendering results. For $N > 2$, we

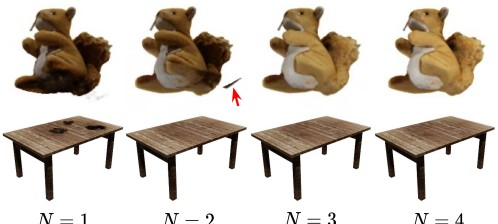

$N = 1$     $N = 2$     $N = 3$     $N = 4$

can observe that the proposed ViFu can recover a clean background and 360° objects from multiple scene observations.

It is noteworthy that, as the number of scenes $N$ increases, the rendered results appear to become brighter. We speculate that this is due to sufficient lighting generally implying less occlusion, which means higher visibility and thus the corresponding parts are fused into the final output with higher weight. Based on this observation, we assume that as the number of scenes and the variety of object

poses increase, the rendering results of objects will be close to those rendered in a 360° spherical lighting environment.

Empirically, we observe that for some objects, artifacts appear when $N = 2$ (red arrow in the figure). We attribute this to the difficulty in accurately segmenting the foreground object if a certain part is in contact with the background part in both two scenes, making it hard to determine whether it belongs to the foreground object or background scene. A simple solution is to expose the part of the common contact when placing objects in the third scene. Although an adhoc placement may achieve the plausible rendering at $N = 2$ (as the cleanser scenes in Fig. 5), we observe that $N = 3$ scenes can achieve reasonable segmentation in most cases and is therefore a recommended choice.

**Impact of variations in object placement.** To validate the robustness of our approach to variations in object placement, we extended our evaluation beyond the 3 original scenes presented in Fig. 1. We created an additional 5 scenes, where objects were randomly placed. We randomly selected 3 from 8 scenes for each fusion experiment. We have the following observations: under the assumptions 3, our method consistently produced satisfactory results. However, some difficulties arise: (1) when the orientations of foreground objects in the three selected scenes are highly repetitive (*e.g.*, bottoms consistently facing downward and thus not observable), artifacts are still present in the rendered regions that lacked sufficient observation. This issue arises because our method relies on fusing information from the available scenes and thus cannot predict unseen part. (2) when foreground objects are placed very closely within a scene, our use of a naive point cloud segmentation approach may potentially fail, leading to misalignment and bad fusion results. Effective segmentation of closely spaced objects typically requires prior knowledge of the objects. Incorporating pre-trained point cloud segmentation models or segmentation masks as additional information can assist in segmenting challenging objects, thereby facilitating successful scene fusion.

## 5 LIMITATIONS AND FUTURE WORK

There are a few limitations that need to be addressed in future work:

First, our method does not explicitly consider the lighting condition. For static background scene, as the lighting conditions are basically the same, reasonable rendering results can be obtained. The above ablation study for light conditions demonstrates that our proposed weighted fusion method can mitigate the impact of certain lighting variations to some extent. However, the rendering results of objects under extreme lighting conditions may still be unsatisfactory (*e.g.*, the fusion result of "construction vehicles" at the top of Fig. 4 shows an abrupt change in appearance, where spot light illumination is used). Incorporating some of the current approaches for disentangling light conditions might be a promising direction for future work.

Second, our fusion method assumes that we can obtain accurate scene segmentation and pose alignment. In most cases, the aforementioned point cloud-based approach can achieve sufficiently accurate segmentation and alignment. However, some challenging scenarios may arise, such as failures in segmentation due to close object placement (as mentioned in Sec. 4.3), or failures in pose alignment due to oversimple object shapes. However, the essence of these problems can all be viewed as fundamentally challenging issues in point cloud segmentation or registration, which has been a longstanding challenging problem in the field of computer vision. For these special cases, using additional masks or richer point cloud features (*e.g.*, color information) might help mitigate the aforementioned challenges.

## 6 CONCLUSION

We have presented ViFu, a method for recovering clean background scene and 360° foreground objects from multiple scene observations. We leverage point cloud-based approaches to achieve background and foreground alignment and use the difference between scenes to obtain a background/foreground segmentation. We propose visibility field, a volumetric representation to quantify the visibility of a scene, and introduce visibility-aware rendering to fuse the more visible parts of multiple scenes. Our experiments on both synthetic and real datasets demonstrate the effectiveness of our approach. While our approach is the first to focus on radiance fields for multiple scenes, there are some remaining issues, such as not considering lighting conditions, which we plan to address in future work.

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
