# OpenReview forum: "ViFu: Visible Part Fusion for Multiple Scene Radiance Fields"
_ICLR.cc/2024/Conference — Submitted to ICLR 2024_

### Official Review · Reviewer_Monn · 2023-10-22

**Soundness:** 2 fair
**Presentation:** 3 good
**Contribution:** 2 fair
**Rating:** 5
**Confidence:** 4

**Summary:**

This paper proposes a method to segment and recover a static, clean background and 360-degree objects from multiple scene observations. The idea is that by observing the same set of objects in various arrangements, parts that are invisible in one scene may become visible in others. By fusing the visible parts from each scene, occlusion-free rendering of both background scene and foreground objects can be achieved. The proposed method first performs objects/background segmentation and alignment based on the point cloud-based methods. It then performs radiance fields fusion, where a visibility field is introduced to quantify the visible information of radiance fields. Last, a visibility-aware rendering is used to obtain clean background and 360-degree object rendering. Experiments were conducted on synthetic and real datasets, and the results demonstrate the effectiveness of the proposed method.

**Strengths:**

This paper studies the under-modeled invisible parts of NeRF and introduces a new setting of complementing the invisible parts by fusing multiple scene information.

This paper introduces a visibility field, a volumetric representation to quantify the visibility of scenes, and proposes novel visibility-aware rendering, which leverages the visibility field to achieve the fusion of visible parts of multiple scenes.

Synthetic and real datasets are created to validate the proposed idea, and the experimental results show the effectiveness of the proposed method.

**Weaknesses:**

One major concern pertains to the practicality of the proposed setup. Reconstructing multiple radiance fields for different variants of a scene, assuming diverse object positions and poses, may be challenging in real-world applications. The implicit assumption of fixed illumination for different radiance fields can limit the practicality of the approach. If we can control the scene, we can reconstruct each object independently rather than using this setup. It would be beneficial to address this concern and discuss potential solutions or scenarios where this setup could be more applicable.

The omission of modeling scene illumination and shadows is a notable limitation. Figure 4 shows that some reconstructed objects have darker or shadowed parts, which raises questions about the fidelity of the reconstructions. I think considering illumination and shadows in the method is critical to enhance the paper's completeness.

While the paper introduces a new setup, the technical novelty of some aspects, such as the segmentation and visibility parts, is limited. Traditional point cloud registration methods and straightforward visibility calculations may not sufficiently push the boundaries of the field.

The quality of results in real scenes appears to be a concern, with reconstructed objects being described as blurry. It is important to provide further analysis or insights into why this issue occurs and potential strategies for improving the quality of real scene reconstructions.

**Questions:**

- Please kindly justify the proposed setup is practical in real-world applications.
- The limitation of not modeling illumination and shadows.
- Please justify the technical novelty of the method.
- Results in the real-world scene is not good.
Please refer to the Weaknesses part for details.

---

> ### Author Response · Authors · 2023-11-18
>
> We appreciate the reviewer's summary of our work! For each question raised by the reviewer, we provide responses below:
>
> > Please kindly justify the proposed setup is practical in real-world applications.
>
> We have outlined some potential application scenarios for our settings in the **Real-world applications** section in the common responses.
>
> > The limitation of not modeling illumination and shadows.
>
> We acknowledge that modeling illumination is beneficial for complex scenes. However, the core focus of this paper is the fusion strategy for given multiple NeRFs, and modeling illumination is a specific improvement for each NeRF reconstruction. Additionally, as our method is a general blending algorithm for neural fields, replacing NeRF with more advanced methods that consider lighting decomposition is a direct solution for more complex scenes. However, some failed experimental results also indicate the importance of lighting conditions for the final results. Therefore, we believe that eliminating the influence of lighting is a valuable future direction. Please also refer to the **Lighting conditions** section in common responses.
>
> > Please justify the technical novelty of the method.
>
> We believe the novelty of our method lies in the core of the proposed approach: the solution and insights into the fusion of multiple neural fields. In recent developments in the 3D field, neural fields have gradually become a mainstream representation alongside point clouds or meshes. When given many 3D assets, fusing them is crucial for practical applications, such as creating new scenes. For point clouds or meshes, their fusion is straightforward (e.g., combining each point cloud). However, for neural fields, due to their implicit nature, how to fuse them is non-trivial. We recognized this fundamental yet unexplored area and proposed a practical method to address it. Therefore, we believe that, despite the simplicity of our approach, it fills a gap in the important direction of neural field fusion, enhances its potential to become a better universal 3D representation, and provides valuable insights for researchers working with neural fields.
>
> > Results in the real-world scene is not good.
>
> We speculate that the occurrence of blurriness in real scenes may have two main reasons. Firstly, inaccuracies in camera pose prediction for real scenes may lead to inaccurate parts in the reconstructed NeRF, causing errors in the object alignment method based on extracted point clouds, resulting in a blurry fusion result. As discussed in the **Robustness** section of common responses, using alternative alignment methods might improve this aspect. Secondly, for real-world scenes, we model the entire surrounding environment using NeRF, not just the foreground objects and the table but also extending to the distant background. This results in the foreground objects having a relatively small proportion in the overall NeRF model. In our implementation, we use Instant-NGP to encode the entire scene, which could result in insufficient resolution to capture fine details of foreground objects during modeling, leading to blurry outcomes. For this aspect, we believe that improving the overall model resolution, or focusing on modeling the space near the foreground object, may lead to improvements.

---

> > ### Comment · Reviewer_Monn · 2023-11-22
> >
> > Thanks to the authors for addressing my comment. My primary concerns are "the practicality of the proposed setup" and the "limitation of not modeling illumination and shadows". Other reviewers also raise similar concerns. However, the authors' responses do not resolve my major concerns. I would keep my initial rating.

---

> > > ### Author Response · Authors · 2023-11-23
> > >
> > > We appreciate the reviewer's feedback.
> > > - Regarding practicality, we believe that the examples mentioned in the common responses, such as indoor mobile robots or wearable visual AI, represent feasible practical applications of our method.
> > > - Regarding lighting conditions, we created scenes with highly reflective foreground objects/background and validated our method under both unchanged and varying lighting conditions. Please kindly refer to Fig. 4 and the light condition section of the ablation study in our updated paper for details. While we agree that modeling illumination and shadows for complex lighting conditions or glossy objects may be a better solution and a promising direction, we believe that, as indicated by our experimental results, our current simple approach can still yield acceptable results in such challenging scenarios.

---

### Official Review · Reviewer_hrRk · 2023-10-29

**Soundness:** 2 fair
**Presentation:** 3 good
**Contribution:** 2 fair
**Rating:** 5
**Confidence:** 4

**Summary:**

The paper tackles the problem of reconstructing and rendering a scene in a compositional manner (both background and multiple objects) given multiple videos of the same scene with rearranged objects. It reconstructs each video individually with NeRF, then uses classic methods (point could alignment, DBSCAN) to register and segment each component. The innovation is a visibility-based compositing method, which composes neural fields based on their visibility (how well a point can be reached by rays from all cameras) in each video.

It shows good qualitative results on a synthetic dataset and a real one.

**Strengths:**

- The writing is clear and easy to follow. The assumptions are discussed thoroughly.
- Multi-scene registration and reconstruction is a practical problem to solve.
- The visibility-based fusion method is sound .

**Weaknesses:**

**Method**
- Scalability. The paper suggests training a per-video representation and using visibility to compose them. This might not scale well in terms of speed and compute cost if there are many videos (needs MxN NeRFs). Since the geometry/appearance of each NeRF is assumed to be consistent across videos, it seems to make more sense to learn a single representation across all videos for each component (needs M NeRFs).


**Experiments**
- There is no quantitative evaluation, comparison with the closest baseline, or ablation study. I think those are important for a paper.
  - Evaluation: how well can the background and objects be segmented (2D IoU) and reconstructed (PSNR / 3D Chamfer distance)?
  - Baseline: I think the shared NeRFs alternative I mentioned earlier is a valid baseline.
  - Ablation: How well does the method work without the proposed visibility composition? How does the selection strategy affect the result (max selection vs weighted)?

**Setup**
- I like the simple setup the paper used to verify the idea. However, it seems too ideal to be generalized to the other cases, for example,
  - Registration failure (e.g., partial scans, nonrigid objects) as mentioned in the limtation
  - The videos have different illumination conditions as mentioned in the limitation.
  - The appearance of the background changes over time (e.g., adding a table cloth)

**Questions:**

Minor comments
- The term "scene" can be confusing. The underlying scene is the same across all videos, but the paper refers to them as different scenes. This makes the definition of visibility confusing. I think naming them video 1...N would be clearer.
- Fig 3 bottom right subfigure, should the points near the end of the ray be red, as they are not visible from the camera?
- In Eq 4, it might be worth showing the exact equation that controls the peakedness (I assume it is similar to temperature-scaled softmax?)

---

> ### Author Response · Authors · 2023-11-18
>
> We appreciate the reviewer's recognition of our work and the detailed suggestions from various perspectives! Below are our responses:
>
> Scalability. Since our method involves NeRF reconstruction for N captures, we indeed need to reconstruct N NeRFs, rather than MxN. For further discussion on the number of scans and required NeRFs, please refer to the **Baseline methods** section in the common responses.
>
> We agree with the reviewer that quantitative evaluation is crucial.
> - **Evaluation**: As mentioned in the paper, finding a quantitative metric that accurately assesses the rendering results of the reconstructed objects is challenging since we aim for the color of the 360-degree objects to be close to the ambient color rather than the color under specific lighting conditions. However, regarding the accuracy of the reconstructed geometry, we agree with the reviewer that the 3D Chamfer distance is a reasonable metric.
> - **Baseline**: As discussed in the **Baseline methods** section in the common responses, in our early experiments, we attempted to separate foreground objects using a segmentation model and individually reconstruct them. However, this process required a significant amount of manual labor for refinement to achieve satisfactory reconstruction results. Nevertheless, we acknowledge that although labor-intensive, using a straightforward method as a baseline is appropriate, and we plan to include a comparison with this approach if possible.
> - **Ablation**: We appreciate the reviewer's suggestions. We will incorporate experiments comparing our proposed visibility-based fusion and naive fusion approaches, such as taking the average.
>
> Regarding setup, please refer to the **Robustness** and **Lightning conditions** sections in the common responses. For the issue of background changing over time, our current method indeed cannot address this, as the current segmentation algorithm assumes that the background remains unchanged across scenes to achieve foreground/background segmentation. However, we agree that this is an important point for practical applications, and improving the generalization of the segmentation step is a feasible direction.
>
> ---
> Minor comments:
> - We appreciate the reviewer's suggestions. We agree that changing "scene" to "video" or "object configurations" might be clearer.
> - Yes, they should be in red. We will correct that.
> - Eq.4 is the specific formula for calculating the blending weight based on the computed visibility. What do you mean by the "exact equation"? Yes, it is similar to a temperature-scaled softmax.

---

> ### Author Response · Authors · 2023-11-23
>
> We have incorporated new experiments on illumination conditions based on your feedback. We created scenes with highly reflective foreground objects/background and validated our method under both unchanged and varying lighting conditions. Please kindly refer to Fig. 4 and the light condition section of the ablation study in our updated paper for details.

---

### Official Review · Reviewer_etRq · 2023-10-30

**Soundness:** 3 good
**Presentation:** 3 good
**Contribution:** 3 good
**Rating:** 6
**Confidence:** 3

**Summary:**

This paper study an interesting problem in novel view synthesis or compositional scene modeling where the some part of the scene is not visible due to occlusion. This paper propose to capture multiple scenes of the same background and same set of foreground objects with different configurations (poses, lightings) and train different NeRF of each scene configuration. Then they propose to align the scene with the point cloud extracted from a trained NeRF by assuming that background have enough overlap and different scene configuration can be aligned based on the extracted point cloud. With the alignment and the proposed visibility field, they can rendering clean background and 360-degree objects.

**Strengths:**

1. This paper study an interesting problem in novel view synthesis or compositional scene modeling where the some part of the scene is not visible due to occlusion which is largely ignored in previous work.
2. The paper proposed to capture/scan the scene with different configurations (same background and same foreground objects but with different poses). Each capture is used to train a NeRF to extract point clouds which are used for alignment between different scenes configurations. I think this is reasonable as the invisible part will never be reconstructed unless it becomes visible.
3. The fusion of different nerf based on visibility field is also interesting.
4. Experiments results show the effectiveness of the proposed method.

**Weaknesses:**

1. Two assumptions seems reasonable but far from reality? For example, if a car on top of the table is never
2. The alignment relies on the extracted point cloud from NeRF. In the simple scenarios shown in the paper, NeRF can do a very good job to extract point cloud, but what if NeRF's point cloud is noisy when there is reflection or textless region?
3. The object segmentation and clustering is heuristic and looks like not very robust. The segmentation of object relies on the clustering algorithm based only point cloud? While I think this simple method works just fine in the simple scenarios studied in the paper, I don't think it will still work in a more complex scene.

**Questions:**

1. Why not use some semantic segmentation method to separate the background and foreground and then to foreground matching of each objects? This seems even easier and straightforward. Or maybe just use the object feature to do matching?
2. If you have control the capturing process, why not just pre capture the scene with only background? And similarly capture the object one by one. I think capturing the scene like this dose not take very much work compared to capturing multiple configurations proposed in the paper.
3. In object clustering, do you need to predefined the number of objects.
4. How to obtain a clean background is not very clear to me? Do you mean if a part is less occlusion, then it belongs to the background?

---

> ### Author Response · Authors · 2023-11-18
>
> We sincerely appreciate the reviewer for finding our work interesting! Thank you for providing valuable suggestions and questions. Below are our responses:
> - Regarding assumptions, please refer to the sections on **Experimental setup** and **Real-world applications** in the common responses.
> - Regarding the robustness of point cloud-based computations, please refer to the **Robustness** section in the common responses.
> - Regarding alternative methods based on segmentation, please refer to the discussion on **Baseline methods** in the common responses.
>
> > In object clustering, do you need to predefined the number of objects
>
> Yes, for robustness, our current method requires specifying the number of objects M. After obtaining the set of foreground point clouds through distance thresholding, we use DBSCAN to obtain multiple point cloud clusters. However, these clusters may include those consisting entirely of noise formed by a small number of points. Therefore, we select the top-M clusters based on the number of points contained in each cluster to obtain the point clouds of each foreground object.
>
> > How to obtain a clean background
>
> In this work, we define the term “background” as the part that remains unchanged across different captures.. We assume that each part of the background (e.g., a part of the table surface) can be observed in at least one capture. For example, an area with no object occlusion in the first capture (no objects above it, no occlusion), but is occluded by objects in subsequent captures (occlusion), our algorithm selects its scene from the first capture as the background. For all areas, we determine the clean background by comparing visibilities, selecting the one with higher visibility.

---

### Official Review · Reviewer_bZoV · 2023-11-03

**Soundness:** 3 good
**Presentation:** 3 good
**Contribution:** 2 fair
**Rating:** 5
**Confidence:** 4

**Summary:**

The paper proposes a method to improve the reconstruction of NeRF models by fusing the visible parts of an objects captured under different scene setting. In order to achieve the goal, it introduces a visibility field which records the visible information of radiance field, and use it to guide the fusion of input images from different scene. The proposed method is evaluated on synthetic and real world captured objects. An ablation study is included to evaluate the sensitivity of proposed method under different lighting, parameter setting, number of input scene, and variation of object placement.

**Strengths:**

The proposed method is simple and easy to follow. It also demonstrated some good results to reconstruct objects with 360 degree visibility.

**Weaknesses:**

The proposed method are only evaluated using "toy" examples. First, the scene captured are all under very simple lighting condition, without any directional light sources, and/or object shadows/reflections. The reconstruction examples are also very simple in its appearance which the observations of the objects do not have any view dependent variations, e.g. specular highlight and/or glossy surface. The different scene setting are also very simple, which each objects can be separated easily. I would consider all the scene setting and examples presented in the paper are well controlled and artificial. Second, in term of technical contribution, once the input images of an object are segmented and are grouped together, we can directly reconstruct the object NeRF model using self calibration method to obtain the relative camera position of each input images. What are the benefits to include the additional visibility field? Besides, I also feel that the proposed method still require very dense observations in order to achieve the high quality reconstruction. Third, the ablation study, except for the evaluation about lighting, I do not think the others are necessary. Instead, I would hope to see deeper analyses on how the proposed method can achieve self-calibration on lighting to resolve that concerns mentioned above about "toy" examples. Note that, although I agree the ablation study on lighting is necessary, but I also do not agree that the current ablation study on variation of lighting is good enough. It is also too simple and does not reflect the real world scenarios as discussed above. Lastly, another limitation is that the proposed method cannot handle dynamic objects. If all the objects are static, there are really not much technical challenges, and static objects can be easily registered.

**Questions:**

Please try to address my comments above.

---

> ### Author Response · Authors · 2023-11-18
>
> We appreciate the reviewer's feedback on our work and the many insightful questions! Below, we provide responses to each of them:
> - Lighting conditions. We utilized directional light in both synthetic dataset and real captures, introducing some level of view-dependent variation. However, we agree with the reviewer's perspective on the importance of validating objects with significant appearance changes across different scenes. We plan to augment our experiments to include more challenging scenarios, such as extreme lighting conditions or glossy objects. Please also refer to the discussion on **Lighting conditions** in the common responses.
> - Regarding the discussion on why we do not use segmented images for reconstruction, please refer to the **Baseline method** section in the common responses.
> - Dense observations. For real captures, as mentioned in Sec. 4.1, we capture a video by circling around the scene, and from that, we extract 60-80 frames for reconstruction. This aligns with the typical setup for 3D reconstruction.
> - As mentioned above, we will add a discussion on lighting conditions in the ablation study.
> - We acknowledge that the current method cannot directly handle dynamic objects. However, considering that ViFu is a general fusion method for neural fields, combining it with some ideas from dynamic scene modeling, such as a deformation field (e.g., introduced in D-NeRF, https://arxiv.org/abs/2011.13961), could be a feasible direction.

---

> > ### Comment · Reviewer_bZoV · 2023-11-21
> >
> > Thanks the authors for the feedback. Although the rebuttal addressed my comments to some extent, I am not fully convinced, since my major criticism is the "toy" examples presented in the papers. I was expecting to see some new examples of challenging scenes with more severe lighting changes, and more thoughtful comparisons with the "baseline" methods. I cannot raise my score based on the current rebuttal and submitted materials.

---

> > > ### Author Response · Authors · 2023-11-23
> > >
> > > We appreciate the reviewer's response. We created scenes with highly reflective foreground objects/background and validated our method under both unchanged and varying lighting conditions. Please kindly refer to Fig. 4 and the light condition section of the ablation study in our updated paper for details.

---

### Official Review · Reviewer_p7zS · 2023-11-03

**Soundness:** 3 good
**Presentation:** 3 good
**Contribution:** 2 fair
**Rating:** 6
**Confidence:** 4

**Summary:**

This paper aims to perform a "piecewise" fusion of NeRFs so that a static background and multiple objects can be separately rendered from arbitrary viewing directions. A scalar volumetric visibility field R^3 -> [0, 1] is proposed, to facilitate the visible part fusion. It assumes all surfaces of all objects are visible somewhere and that accurate pose alignment can be found for all objects. Qualitative results are shown on a synthetic dataset of objects rendered on a flat tabletop surface, as well as a real dataset in a similar setting.

**Strengths:**

Well motivated. Although the assumptions seem too strong for in-the-wild data in its current form, I can see further development happening, to jointly optimize pose, perform completion, and be able to handle different lighting conditions.

**Weaknesses:**

No quantitative experiment. Some analysis of the number of images or viewpoints needed would have been interesting. The paper says the method consistently produced satisfactory results, but in some sense, that is guaranteed to happen given satisfactory input. It would have been better if there was more we could learn from the paper experimentally, including failure cases. My interpretation is that the paper has two take-home messages: 1. visible part fusion is a problem that needs more attention. 2. we can define volumetric density functions based on visibility to serve as a blending weight function, assuming we have everything else needed. On second thought, while the presentation is great, I wish I could see more.

**Questions:**

Perhaps consider citing FiG-NeRF: "separating foreground objects from their varying backgrounds" (instead of the same static background).

What if the background isn't flat?

How robust is this to noise in point-cloud registration?

---

> ### Author Response · Authors · 2023-11-18
>
> We thank the reviewer for summarizing our work! The two take-home messages mentioned by the reviewer indeed are our main focuses and contributions. We provide specific responses below:
>
> Failure cases. We acknowledge the importance of failure cases. In our method, failure cases primarily arise from two aspects: (1) failure in object pose registration and (2) significant appearance discrepancies between different captures. We have added discussions on these two aspects in the section of **Robustness** and **Lightning conditions** in our common responses.
>
> > Consider citing FiG-NeRF
>
> We appreciate the reviewer's suggestions. We find its motivation similar to ours, and we will add it in the citation.
>
> > What if the background isn't flat?
>
> There shouldn't be any issues. In our assumption, if a part remains unchanged in all scenes, we consider it as part of the background. For example, if a keyboard on a desk doesn't move in each capture, it is considered part of the background, and in this case the separated background will include both the desk and the keyboard. A flat background is not our assumption. (If time permits, we would like to append experimental results under non-flat background conditions.)
>
> > How robust is this to noise in point-cloud registration?
>
> Please refer to our discussion on **Robustness** in the common responses.

---

> ### Comment · Reviewer_p7zS · 2023-11-23
>
> I appreciate the authors' response. The major concern from other reviewers was the simplicity of the examples, which I did not find problematic. However, the lack of quantitative evaluation, as noted in the weaknesses section, remains unaddressed. I would have preferred to see some indication of potential experimental validation results in the future, but the response was not convincing. Despite the paper's clean presentation and strong motivation, I choose to maintain my initial rating. I am inclined towards borderline, which unfortunately, I cannot select as an option.

---

### Author Response · Authors · 2023-11-18
**Common responses (1/N)**

We would like to thank the reviewers for their careful and constructive feedback! For some common concerns, we summarized our responses below. We will also provide individual responses to each reviewer's specific questions.
# Experimental setup
We acknowledge that our current experimental setup may lean towards the ideal, but we believe such a configuration maximizes the elimination of potential disturbances, thereby validating the effectiveness of the proposed method for NeRF fusion. For potentially more challenging scenarios, we believe that certain modifications can mitigate possible issues (discussed in the upcoming Robustness section).

Reviewing our experimental setup, we aim to achieve the reconstruction of foreground objects and background without occlusion with a minimal number of scans for the background and several different arrangements of foreground objects. This is inherently a challenging setup. While we could perform separate scans/segmentations/reconstructions for the background or objects individually, we prefer these reconstructions to be accomplished simultaneously in an automatic manner. We consider this direction to be meaningful, especially considering the potential future demand for scanning a large number of objects, potentially in the hundreds or thousands, where individually processing each object would be labor-intensive. From this perspective, we designed the current experiment setup to scan multiple objects simultaneously.

Please also refer to our discussion in the upcoming Baseline methods section.

# Real-world applications
Firstly, due to the capability of our method to simultaneously scan multiple objects, it is a natural application to scan a large number of objects and the background.

Additionally, although becoming more specific, our envisioned application scenarios align well with everyday visual AI applications, such as indoor mobile robots or wearable visual AI devices like AI glasses.
- Our problem setup is well-suited for such applications. Firstly, regular acquisition of camera images becomes straightforward due to the routine movement in indoor spaces. Secondly, natural placements of objects in various positions and orientations occur as part of people's daily activities. Moreover, in indoor environments with many objects, capturing each foreground object or clean background separately may not be easily achievable. Therefore, shared capture of multiple objects/backgrounds is considered a direction to reduce manual labor.
- Our research objective is closely related to such applications. For indoor mobile robots or wearable visual AI devices, creating a simulation environment closely resembling the real world is crucial for system learning. For example, when considering a robot's task of tidying up a room, we want to generate foreground objects in different positions and orientations on a table in a simulated environment for the robot learning. In such applications, the separation and modeling of 360-degree foreground objects and background become crucial.
- While we acknowledge that implementing our method in the mentioned applications may require some modifications, we believe that our problem formulation and research objective have value for these application scenarios and can provide insights to the research community.

Moreover, from the technological perspective, the fusion of multiple NeRF representations is an important and yet under-explored technique. Our contributions in this work are fundamental not only for applications but also for the development of NeRF research.

**(To be continued)**

---

### Author Response · Authors · 2023-11-18
**Common responses (2/N)**

**(Continuing from common responses 1/N)**
# Baseline methods
Reviewers mentioned that segmenting or individually scanning objects to model them independently is a straightforward and reasonable baseline method. We consider two baseline methods and explain why we do not prefer these methods.
- **Baseline 1**: Scanning the background and each object separately, and reconstructing them individually.
- **Baseline 2**: Employing a similar capture method as ours (i.e., capturing videos under N different pose configurations), using a pretrained segmentation model to segment the background and each foreground, and reconstructing them separately based on the segmented images.

Firstly, we do not use **Baseline 1** because of its higher cost, which contradicts our goal of simultaneously reconstructing multiple objects with minimal manual intervention. Specifically, considering the scenario of reconstructing M foreground objects and 1 background in N different configurations, the comparisons between our method and the two baseline methods regarding (1) the number of required captures and (2) the number of NeRFs needed are as follows:
- **Ours**: N captures, N NeRFs
- **Baseline 1**: (M+1)N captures, M+1 NeRFs
- **Baseline 2**: N captures, M+1 NeRFs

In our experiments, it is often sufficient to capture three different configurations to observe all parts of an object (i.e., N=3). On the other hand, M can be relatively large; in our experiments, the maximum value of M is set to 5. However, in scenarios involving scanning a large number of objects, M could potentially be in the hundreds or thousands (although this would require a more sophisticated method). Moreover, if a pretrained segmentation model is used to obtain foreground objects, this method may entail a significant amount of manual labor, similar to the challenges described in Baseline 2 below. Given this, we believe that Baseline 1 requires more human effort, which is why we opted for the shared capture approach for multiple objects.

Secondly, we refrain from using **Baseline 2** primarily due to potential issues introduced by the pretrained segmentation model used during segmentation. In our initial experiments, we also utilized a pretrained segmentation model to obtain foreground objects and observed some typical failure cases: (1) Over-segmentation due to excessively strong semantic information. For example, attempting to segment a plastic bottle resulted in the bottle cap and body being segmented as two different instances. (2) Segmentation failures due to occlusion between objects. While these failures can be mitigated through manual refinement or annotation, it significantly increases the manual labor involved. Another perspective is that, from a practical standpoint, scanning unique objects may hold more value (e.g., obtaining a 3D model of a plastic bottle may be easier to find online than to re-scan). Unique objects like these may not be well-represented in datasets used to train segmentation models, potentially leading to suboptimal segmentation results. For instance, the toy chicken racer in Fig. 1 is detected as a teddy bear with low confidence by Detectron2 (https://github.com/facebookresearch/detectron2), and the segmentation result shows noisy edges. These considerations motivated our use of the proposed segmentation method entirely based on geometry information. While current classical point cloud-based methods may lack robustness, we believe that exploring this direction, complementary to semantic-based methods, holds value.

**(Continued)**

---

### Author Response · Authors · 2023-11-18
**Common responses 3/N**

**(Continuing from common responses 2/N)**

# Robustness
Reviewers raised concerns regarding the robustness of our classical point cloud-based approach for foreground/background segmentation and alignment. Firstly, we performed filtering operations (i.e., statistical outlier removal and radius outlier removal) on the point cloud extracted from NeRF, significantly reducing the impact of noisy point clouds. For subsequent operations, we have the following observations: our point cloud-based method proves to be robust for scene alignment and foreground/background segmentation; however, for object pose alignment, failures may occur due to noisy or inaccurate point clouds. A detailed analysis is provided below:
- **Scene pose registration**: As mentioned in the supplemental material, relying solely on point clouds for inter-scene registration can sometimes lead to failures. Therefore, in addition to point cloud information, we utilize RGB images and accomplish scene alignment through feature matching. Specifically, we employ COLMAP for registration, making this step relatively robust and effective.
- **Foreground/background segmentation**: The aforementioned scene pose registration is relatively accurate, resulting in an accurate model of the clean background. Foreground point cloud is obtained through thresholding based on the distance between foreground and background point clouds. This simple thresholding operation is robust against point cloud noise because minor noise in points located a certain distance from the background does not significantly impact the segmentation results.
- **Object pose registration**: We acknowledge that this step demands high precision in point clouds and may face challenges with noisy or inaccurate point clouds. However, registration of NeRFs across multiple scenes is still an unexplored area. A recent work nerf2nerf (https://arxiv.org/abs/2211.01600), has made an initial attempt to address this issue, but due to its inherent difficulty, the method relies on manual annotation of pairs of keypoints. Since our primary focus is on how to fuse NeRFs of multiple foreground objects/backgrounds, for the sake of simplicity in the entire workflow, we utilized a straightforward but fully automated traditional point cloud-based method for object registration, eliminating the need for manual annotations. We believe that for relatively simple scenes, our approach can be considered a minimal, yet automatic and effective method. For more challenging and complex scenes, replacing the object pose registration part in our pipeline with nerf2nerf (though it requires manual annotation) could be a feasible alternative.

# Lightning conditions
- Directional light was used in creating the dataset. As mentioned in Sec. 4.1, we utilized area light, directional light, or a combination of both for synthetic dataset creation. In real-world scenes, the environment was illuminated by multiple spotlights.
- We acknowledge the reviewer's primary concerns regarding the potential impact on fusion results when there is significant variation in the appearance of objects across different scenes, due to lighting or surface material issues.  In some cases, such as when certain parts of objects lack sufficient illumination in all scenes, there may be discontinuous appearance changes in the synthesized results (as observed in the construction vehicle in Fig. 4). We concur with the reviewer's perspective and recognize the importance of addressing this aspect. We plan to supplement our work with additional experiments and discussions specifically focusing on situations where appearance changes are substantial (e.g., diverse lighting conditions, glossy objects).
- We also want to emphasize that regarding the 360-degree reconstruction of foreground objects, our typical objective is to obtain the object's appearance under uniform lighting conditions (i.e., diffuse color), rather than under specific lighting conditions. As mentioned in the ablation study concerning the number of scenes N, our method tends to produce colors resembling albedo as N increases. Even for objects with strong view-dependent appearances, our approach can neutralize these view-dependent terms through multi-scene fusion, leading to reasonable results.
- As mentioned in the limitations section, an alternative viable approach could involve replacing part of NeRF with a model that decomposes appearance, such as Ref-NeRF. Fusion of the diffuse color from these models is feasible, given that ViFu is a general fusion method for neural fields.
- Regarding the shadows generated between objects, our method also possesses the capability to remove shadows, as demonstrated in a manner similar to the ablation study presented in the supplemental material.

---

### Author Response · Authors · 2023-11-23
**Paper updates**

We added experiments with highly reflective foreground objects/background, validating our method under both consistent and varying lighting conditions. The results are presented in Fig. 4 and the light conditions section of the ablation study in the updated paper. The modified portions are highlighted in red.

---

### Meta-Review · Area_Chair_KJjF · 2023-12-06

**Metareview:**

This paper presents a method for fusing the visible parts of an object in NeRFs so that distinct objects can be individually rendered from arbitrary viewpoints. To this end, it introduces a visibility field, which records the visibility information of the radiance field, and uses it to guide the fusion of different scenes. The proposed method is evaluated using synthetic and real-world scenes, and the results are convincing.

The major strengths of the paper are:
(1) The idea of fusing different NeRFs based on the visibility field is somehow interesting.
(2) Experimental results showcase the effectiveness of the proposed method.

On the other hand, the weaknesses are
(1) The proposed object segmentation and clustering are rather heuristic.
(2) The setting is unrealistic, and the employed scenes are too simple.

Overall, while the reviewers appreciate the work and quality result, there remains a concern about the practicality of the setting and robustness of the proposed method. The authors' rebuttal has addressed these issues to some extent; however, unfortunately, it did not fully eliminate the concerns. As a result, we reach this final recommendation.

**Justification For Why Not Higher Score:**

Although the reviewers and AC appreciated the quality of the result, there were remaining concerns about the practicality of the setting and the robustness of the proposed method, particularly in segmentation and clustering.

**Justification For Why Not Lower Score:**

N/A

---

### Decision · Program_Chairs · 2024-01-16

Reject